# New Insight into Ornamental Applications of Cannabis: Perspectives and Challenges

**DOI:** 10.3390/plants11182383

**Published:** 2022-09-13

**Authors:** Mohsen Hesami, Marco Pepe, Austin Baiton, Seyed Alireza Salami, Andrew Maxwell Phineas Jones

**Affiliations:** 1Department of Plant Agriculture, University of Guelph, Guelph, ON N1G 2W1, Canada; 2Department of Horticultural Sciences, Faculty of Agricultural Science and Engineering, University of Tehran, Karaj 31587-77871, Iran; 3Industrial and Medical Cannabis Research Institute (IMCRI), Tehran 14176-14411, Iran

**Keywords:** hemp, horticulture, potted plant, landscaping, chemotype Ⅴ, legalization, breeding

## Abstract

The characteristic growth habit, abundant green foliage, and aromatic inflorescences of cannabis provide the plant with an ideal profile as an ornamental plant. However, due to legal barriers, the horticulture industry has yet to consider the ornamental relevance of cannabis. To evaluate its suitability for introduction as a new ornamental species, multifaceted commercial criteria were analyzed. Results indicate that ornamental cannabis would be of high value as a potted-plant or in landscaping. However, the readiness timescale for ornamental cannabis completely depends on its legal status. Then, the potential of cannabis chemotype Ⅴ, which is nearly devoid of phytocannabinoids and psychoactive properties, as the foundation for breeding ornamental traits through mutagenesis, somaclonal variation, and genome editing approaches has been highlighted. Ultimately, legalization and breeding for ornamental utility offers boundless opportunities related to economics and executive business branding.

## 1. Introduction

Cannabis (*Cannabis sativa* L.), belonging to the Cannabaceae family, is one of the world’s oldest domesticated crops. This multi-purpose plant is used for food (e.g., oil, seeds, herbal tea), raw fiber (e.g., textiles, ropes, building materials), bioenergy, recreation, and medicine [1] (Figure 1).

Additionally, cannabis remains an important part of several traditional cultures and a feature of ethnological myths [2]. Although there exist several complications when defining cannabis genetics (i.e., species nomenclature), genetic differentiation between drug (marijuana)-type (rich in Δ^9^-tetrahydrocannabinol (THC)) and seed/fiber-type (containing low levels of THC) remains well-founded [3].

Cannabis is a vigorous and fast-growing crop [4]. Due to its variable growth habit, abundant green foliage, and distinguishable leaves [4], cannabis can be an attractive ornamental plant. The short, strongly branched cultivar “Panorama” was bred and commercialized by Iván Bócsa as a Hungarian ornamental crop in the 1980s [5]. This cannabis strain remains one of the only ornamental cultivars recognized by horticulturalists. Furthermore, decorative cannabis bonsai and terrarium products are sold by several retail companies in some countries such as Canada. However, legal restrictions and difficulties differentiating this crop from illegal genotypes has limited its utility as a commonly adopted ornamental plant [5]. Additionally, some in the horticultural industry regard cannabis as a toxicity risk for small children and domestic pets/animals [6]. This reasoning must be re-assessed, since other medicinal plants (e.g., *Euphorbia pulcherrima*, *Catharanthus roseus*, *Digitalis purpurea*, *Papaver somniferum*, *Dieffenbachia seguine*, *Echinopsis pachanoi*, *Nerium oleander*, *Artemisia absinthium*, *Rhododendron ponticum*) are much more psychotropic and/or toxic, yet are extensively cultivated worldwide as high value ornamental crops [6].

In reality, cannabis is much less toxic than many popular house plants and this negative stance related to the toxicity of cannabis is largely a result of extensive propaganda aimed at demonizing the species rather than fact. For instance, handling and consuming *Philodendron* or *Dieffenbachia*, two popular house plants, can lead to allergic reactions due to their production of calcium oxalate [7,8]. Additionally, accidental exposure to *Datura* and *Brugmansia* as two popular house plants results in 20% of the fatal outcomes, being the leading cause of death attributed to plant exposures [9]. In contrast, cannabis in its fresh form has only been reported to cause mild allergic responses due to cannabinoids and pollen [10]. Moreover, the bulk of cannabinoids present in the fresh plant is in their acid form and is not psychoactive. Most allergic responses can occur only after the plant is processed and the compounds convert into their non-acid form [11]. Moreover, the two main allergens of cannabis include cannabinoids found in leaves and/or inflorescence, and pollen grains from male flowers [10,11]. These concerns can be overcome by breeding feminized chemotype V cannabis strains with an undetectable amount of phytocannabinoids (almost zero phytocannabinoids) for producing ornamental cannabis. Since feminized chemotype V varieties produce almost no phytocannabinoids and zero pollen, the allergic effects of cannabis can easily be quelled. Given the more recent liberalization of cannabis regulations around the world, including the ability for individuals to cultivate plants at home in many regions, there is significant potential to develop ornamental or multi-purpose cultivars. Moreover, people lacking negative social pre-conceptions toward cannabis do grow cannabis as featured ornaments.

Breeding new cannabis cultivars with high pharmacological potential has been the focus of many researchers, due to the ever-growing demand for pharmaceutical and recreational cannabis [12,13]. As a result, numerous cannabis breeding methods have been devised and refined over the past few years, giving rise to medicinal-grade cultivars with a variety of chemotypes [14,15]. Ongoing efforts to develop low-THC cultivars with medicinal and industrial importance provide a solid foundation to develop ornamental cannabis strains with reduced risk of diversion. Additionally, several morphological markers (e.g., variegated foliage) represent recognizable characteristics that distinguish protected cannabis varieties from those unprotected, allowing specific cultivars with valuable ornamental characteristics to be identified [16]. Despite the ornamental feasibility of cannabis, its suitability for the ornamental industry remains largely unrecognized, and studies tend to focus on botany [17], cultivation [18,19], propagation [20,21], and agricultural [4] aspects of cannabis. 

Ornamental plants are critical to the beautification of both interior and exterior settings. These scenic atmospheres can prompt emotional responses that increase demand for certain plants and drive the horticultural industry [22]. Commercial trade of ornamental plants represents a globally growing component of the horticultural industry that currently contributes billions of dollars per year and is predicted to increase by 6–9% annually [23]. Based on consumer preference, market potential, and aesthetic significance [24], introduction of new and exotic alternatives has become the primary means to meet rising demands for ornamental plants [25]. However, several requirements and criteria must be carefully considered when assessing the suitability of new ornamental plants before introducing them to market.

By assessing multifaceted criteria standardized by the industry, this study set out to evaluate the suitability of cannabis for introduction as an ornamental horticultural specialty crop through an unbiased lens. Related opportunities and challenges of introducing such plants to different facets of the ornamental plant sector are then presented and highlighted. Finally, promising approaches to breeding ornamental traits into cannabis have been discussed. Ultimately, this work summarizes various ways to develop and produce novelty cannabis cultivars with marketable ornamental features of high value to the horticultural industry. 

## 2. Cannabis as a New Ornamental Plant

Introducing a new ornamental crop is a complex process due to the diverse range of selection criteria that can be applied. Criteria for bedding/garden/landscape plants focus on growth habit, vegetative and reproductive period, plant height, concomitant survival rate, and tolerances to biotic and abiotic stress [26]. For potted plants, requirements for consideration include leaf shape, leaf length, leaf texture and shine, stem rigidity and appearance, and flower/inflorescence color and size [27]. In the current study, we aimed to provide an unbiased assessment of cannabis’ ornamental potential by utilizing the procedure developed by Krigas et al. [28], which accounts for all subsectors of the ornamental industry (e.g., cut-flowers, pot plants, bedding plants). This evaluation method contains three levels: (i) potential in the ornamental sectors, (ii) sustainable exploitation feasibility, and (iii) readiness timescale for new value chain creation (Figure 2a).

At the first level (Figure 2b), a point-scoring approach with twenty sector-specific markers (frost hardiness, shade tolerance, wild collections, altitudinal range, blooming period, compactness, environmental tolerance, height, breeding possibility, botanical holidays, cut flower eligibility, salt tolerance, impressive flowers, leaf color, plant symmetry, seasonal phenotypic changes, attractiveness of leaf shape, eligibility as foliage plant, e-trade over the internet, and shining of leaf texture) was used to evaluate the general ornamental potential of cannabis based on the available literature [4,29,30,31,32,33,34,35,36,37,38,39]. Then, the relative percentage of the ornamental potential of cannabis was calculated based on the following equation:$$\mathrm{Ornamental}\;\mathrm{potential}\;\mathrm{of}\;\mathrm{cannabis}=\left(\frac{\;\sum \mathrm{Score}\;\mathrm{of}\;\mathrm{each}\;\mathrm{attribute}\;}{\sum \mathrm{Maximum}\;\mathrm{score}\;\mathrm{of}\;\mathrm{each}\;\mathrm{attribute}}\right)\times 100$$

Results specify that cannabis has an ornamental potential score of 78.33% compared to a standard 67.5% score [28]. This indicates a 10.83% higher ornamental potential relative to the standard for introduction.

To evaluate the ornamental potential of cannabis as a potted plant, the following equation was used:Suitability as potted plant
= 0.75 × (the score for leaf color
+ the score for plant symmetry
+ the score for the blooming period
+ the score for impressive flowers
+ 0.25
× (the score for existing prices in the electronic trade
+ the score for seasonal phenotypic changes
+ the score for compactness of form
+ the score for height
+ the score for the attractiveness of leaf)

Results indicate that cannabis shows a high suitability (77.45%) for consideration as a potted plant.

Additionally, the following equation was used to assess the suitability of cannabis for landscaping.
Suitability for landscaping
= 0.75 × (the score for water demand)
+ the score for environmental tolerance
+ the score for altitudinal range
+ the score for frost hardiness
+ the score for seasonal phenotypic changes
+ the score for height + the score for plant symmetry
+ the score for compactness of form
+ the score for blooming period)
+ 0.25 × (the score for existing prices in the electronic trade
+ the score for leaf color
+ the score for the attractiveness of leaf shape
+ the score for impressive flowers

Results exemplify cannabis’ high suitability (87.63%) for landscaping. Due to their well-documented tolerance to heavy metal stress, industrial hemp can be employed for phytoremediation in radioactive and contaminated soils [40,41]. This, along with its high potential for landscape planting, makes ornamental cannabis an ideal and valuable candidate for landscapes that surround nuclear, oil, metal, and other related factory lots. Thus, landscapes in industrial areas can be improved by including ornamental cannabis to reduce soil contaminants, while providing pleasant aesthetics and distinct aromas.

Based on this assessment, cannabis has great ornamental potential. However, the score for existing prices in the trade depends on the legal status of cannabis. Hence, legalization can further increase the value of cannabis as an ornamental crop. An internet search identified select retail companies that make decorative cannabis available to customers of specific regions, identifying an existing market for decorative cannabis bonsai and terrarium products. However, as previously mentioned, legal obstacles associated with cannabis cultivation continue to present major bottlenecks for all aspects of the cannabis industry [42]. This barrier is difficult to overcome, and would require fundamental restructuring of regulations on both national and international levels [14]. Thanks to recent advances in molecular markers and genetic engineering methods (e.g., CRISPR), it is possible to detect and/or produce cannabis with very low levels of THC [43,44]. Though there could still be complications differentiating ornamental from medicinal varieties, cannabis cultivars that produce low cannabinoid levels might help the ornamental industry to overcome legal barriers and prevent unwanted diversion. Additionally, some other scores (e.g., the score for impressive flowers) can be enhanced with plant breeding methods, which will later be discussed in more detail (see the section “breeding and biotechnological methods for producing ornamental cannabis”).

Arguably, certain sector-specific markers can only be evaluated subjectively (impressive flowers, cut flower eligibility, attractiveness of leaf shape), whereas others remain dependent on cultivar (seasonal phenotypic changes, leaf colour, blooming period) (Figure 2b). This has extensive implications when evaluating cannabis using such methods, since there are many different cultivars of cannabis, each with different phenotypic properties and growth characteristics. Likewise, while many people may enjoy the aesthetics of cannabis flowers, others would strongly dislike it, largely due to its history and cultural associations. To avoid a biased evaluation, we considered only the minimum eligibility scores for these subjective attributes. Still, results demonstrate high suitability scores overall, regardless of using minimum eligibility scores, and it is possible that cannabis might have scored higher based on evaluator preference, or cultivar specificities.

At the second level (Figure 2c), a partial scoring of twelve markers (existing cultivations, cultivation needs, existing cultivation protocols, water demand, known propagation, seed germination success, vegetative propagation success, distribution in national regions, phytogeographic regions, ex-situ conservation in institutions, protection status, threat category, and commercial products on market) was employed to evaluate the sustainable exploitation feasibility of cannabis in the ornamental industry [10,30,33,37,45,46,47,48,49,50,51,52,53,54,55,56,57,58].

Using these markers, the relative percentage of the sustainable exploitation feasibility of cannabis was calculated based on the following equation:$$\begin{array}{c} \mathrm{Sustainable}\;\mathrm{exploitation}\;\mathrm{feasibility}\;\mathrm{of}\;\mathrm{cannabis}\\ =\left(\frac{\;\sum \mathrm{Score}\;\mathrm{of}\;\mathrm{each}\;\mathrm{attribute}}{\sum \mathrm{Maximum}\;\mathrm{score}\;\mathrm{of}\;\mathrm{each}\;\mathrm{attribute}}\right)\times 100 \end{array}$$

A very high sustainable exploitation feasibility (80.56%) for cannabis in the ornamental industry was achieved due to the amount of available information related to botany [4], biology [30], agriculture [42,59], propagation [60], and cultivation [20] of cannabis.

At the third level, general analyses of SWOT (strengths, weaknesses, opportunities, and threats) and GAP are fundamental prerequisites in determining the readiness timescale for new value chain creation (Figure 2a). Introducing ornamental cannabis has several potential economic (e.g., creating jobs, improving the ornamental industry) and environmental benefits. For instance, we previously mentioned several environmental benefits of introducing cannabis to the landscape industry (e.g., reducing the level of nitrate in groundwaters, decreasing disease and pests in other plants, absorbing more carbon [42], improving soil conditions [61], and reclaiming heavy metal contaminated soils). These benefits outweigh the value of many currently available ornamental landscaping plants such as *Sambucus racemosa*, *Rhus typhina*, *Deutzia gracilis*, *Melianthus major*, and *Salix integra*.

Since numerous attributes at the third level (e.g., estimated exploitation of distribution channels, estimated difficulty for value chain creation, estimated attraction of new producers-retailers, possibility to overcome legal restrictions) completely depend on the legal status of cannabis (Figure 2d), decriminalization is critical before introducing cannabis into ornamental horticulture industries. Although cannabis remains illegal in many countries, rescheduling of cannabis by the United Nations [62] has provided the preliminary framework necessary for associated countries to reconsider their laws and regulations related to cannabis products. Alternatively, countries that have already legalized cannabis (e.g., Canada, The Netherlands, Malta, and Uruguay; industrial hemp is cultivated legally in most European countries, for varieties registered in the European catalog and their THC levels are <0.2%) could initiate their ornamental cannabis industries in a short time, with minimal adjustments to policies.

## 3. Breeding and Biotechnological Methods for Producing Ornamental Cannabis

Applications of sequencing technologies and molecular genetic markers has resulted in boundless advances related to cannabis breeding for desirable horticultural characteristics [63]. Various molecular genetic markers have also been employed in the cannabis field to analyze genetic variations, sex determination, chemotype inheritance, and genetic mapping (reviewed by Hesami et al. [30]). For instance, Mandolino and Carboni [44] employed molecular markers to study chemotype inheritance in cannabis. They discovered a chemotype with an undetectable amount of phytocannabinoids (almost zero phytocannabinoids) and classified it as chemotype Ⅴ [44]. Johnson and Wallace [64] employed genotyping by sequencing (GBS) to evaluate chemotype inheritance in cannabis. They also found several accessions with no detectable phytocannabinoids (chemotype Ⅴ). Since this chemotype of cannabis produces virtually no phytocannabinoids, it can overcome some legislation related to cannabis. Therefore, chemotype V represents an ideal candidate for breeding and producing ornamental cannabis strains.

High-throughput genotyping approaches (e.g., GBS) have also resulted in the detection of single nucleotide polymorphism (SNP) markers, and the construction of a genetic linkage map for cannabis [65]. Such approaches can be useful to identify candidate genes and favorable SNP alleles for different ornamental cannabis breeding objectives such as stress resistance- and flowering-related characters [51]. Another advanced approach, genome-wide association study (GWAS), is useful for understanding complex traits (e.g., plant architecture, leaf character, florogenesis, stress resistance) and identifying related candidate genes [66] for breeding ornamental cannabis. Although project goals can vary among ornamental breeders, the production of novel cultivars with specialty traits and high commercialization potential is the most important theme of these cannabis breeding ventures. The principal economically valuable and commercially important traits for breeding ornamental cannabis (for both potting and landscaping plants) are presented in Figure 3.

Different biotechnological and breeding approaches such as crossing methods, polyploid induction, mutagenesis, in vitro-based breeding methods (e.g., embryo rescue, somaclonal variation), and genome engineering can also be employed to produce ornamental cannabis (Figure 4). There are several exceptional review papers presenting different biotechnological and breeding strategies for cannabis, including crossing [67], domestication [18], polyploid engineering [68], plant tissue culture [60], and genetic engineering [68]. The methods presented in these review papers can also be used for breeding and production of ornamental cannabis. Here, we discuss and highlight select methods such as mutation breeding, somaclonal variation, and genome editing for producing ornamental cannabis varieties.

### 3.1. Mutation Breeding

Detection and selection of mutants with high commercial value is a common method of introducing new cultivars to the ornamental industry [69]. Different types of spontaneous mutations (germline mutations, somatic mutations, and epimutations), either beneficial or deleterious, occur in cannabis just like in other plants. Adamek et al. [70] showed that somatic mutations occur with high frequency in cannabis and that these mutations differ in various parts of individual plants. There are several cannabis genotypes and/or cultivars with valuable ornamental features that were presumably selected based on spontaneous mutations (Figure 5). For instance, genotypes with whorled phyllotaxy, containing three or more leaves at each node, have resulted from spontaneous somatic mutations. In addition, “Ducksfoot” strains were also derived from spontaneous mutation. This strain is of high value as a potted plant due to its webbed leaves and beautiful purple inflorescences. Australian Bastard Cannabis (ABC) is yet another potential mutant for the potted plant industry, bearing a succulent shrub-like shape that contains shiny, small, smooth, and non-serrated leaves. The cultivar “Divina”, introduced by Casano [6], is a mutant with albinism that produces variegated leaves and tissues. This is the only selected mutant introduced as an ornamental cannabis plant.

In addition to spontaneous mutations, induced mutation is considered a robust approach to developing novel cannabis cultivars with valuable ornamental characteristics (e.g., biotic and abiotic stress tolerance, foliage traits, floral traits, growth habit and pattern, photoperiodism) for the horticultural industry [71]. To date, over 728 species-specific ornamental mutant cultivars have been released worldwide [69]. Recent advances in mutagenesis technology (both technical details and ideas) have driven ornamental breeder focus toward induced mutation techniques for introducing new ornamental varieties. Since mutagenesis alters only select traits of elite cultivars [72], this technique can be useful for plants such as cannabis that are asexually propagated. Moreover, the heterozygous nature of cannabis is an advantage allowing a high frequency of induced mutations. Both chemical (alkyl group) and physical (gamma rays and heavy ions) mutagens can be applied to different forms of cannabis, such as in vitro plantlets, floral tissues, leaves, cuttings, seedlings, and seeds [69]. The dosage and exposure time of mutagens directly influences the type and frequency of mutations. Therefore, it is necessary to optimize such procedures to successfully induce mutation [73]. Fuochi et al. [74] employed radiation-based mutagenesis and produced two low-THC cultivars of cannabis (less than 0.17%) with yellow distal leaflets and red petioles. These cannabis cultivars represent exceptional candidates for introduction to the ornamental industry.

The main challenge with mutagenesis is the formation of chimeras, by which mutations occur in a single cell, then develop into cell layers that can emerge as a fine strip on a portion and/or entire organ (e.g., leaf, floret, branch). Isolating chimeric tissue is a critical step in mutagenesis. Thankfully, advanced plant tissue culture techniques allow the separation and establishment of chimeric mutant tissue in their pure form (reviewed by Ibrahim et al. [73] and Datta [69]).

### 3.2. Somaclonal Variation

Somaclonal variation refers to all forms of in vitro culture-mediated phenotypic variation, potentially due to spontaneous mutation, chromosome mosaics, or epigenetic machinery [60]. Somaclonal variations can appear in calli, isolated protoplasts, tissues, and organs as well as in vitro raised plantlets [20]. It is only on occasion that plant tissue culture scientists detect somaclonal variations with commercial value and introduce them as new ornamental cultivars [75]. Since a high frequency of somaclonal variation is required for practical application, this method cannot be routinely employed for developing novel cultivars. However, there are several successful examples of somaclonal variants in the ornamental industry, including the distinct foliar pattern and large canopy of the Dieffenbachia plant [76], biotic stress tolerance of carnations [77], and size and number of flowers per begonia plants [78]. Although there are currently no cannabis cultivars derived by these means, indirect organogenesis can be considered an alternative method for introducing new varieties through somaclonal variation [60,79].

### 3.3. Genome Editing

Functional and structural information of gene regulatory networks are necessary for developing genome editing-based approaches. Although the whole-genome sequence in cannabis has been reported, the heterozygosity and complexity of the cannabis genome, coupled with challenges related to plant regeneration, have slowed the development of genome editing approaches [80]. In fact, reliable and accurate genomic resources are of paramount importance for designing single guide RNAs (sgRNAs) [81]. We have previously reviewed online tools containing cannabis genome resources for designing sgRNAs, the strategies to avoid off-target activities, different methods of gene transformation (e.g., Agrobacterium-, biolistic-, and virus-mediated delivery), and the application of morphogenic genes in cannabis elsewhere [68].

Most CRISPR-based methods rely on double-strand breaks (DSBs), which involves cleaving the double-strand DNA at a target site. With lack of any template, non-homologous end joining (NHEJ) functions to repair cleavages results in small insertions or deletions (indels). In the presence of a template, homology-directed repair (HDR) can induce desired point mutations or indels at target sites [82,83,84,85]. In relation to cannabis, Zhang et al. [86] successfully employed the NHEJ-based CRISPR/Cas9 method to knock out the phytoene desaturase gene (CsPDS1). DNA sequence encoding and Cas9 were introduced to immature zygotic cannabis embryos through Agrobacterium-mediated gene transformation, creating albino plantlets [86]. Such approaches may also be employed to introduce new varieties of cannabis with valuable ornamental traits. Additionally, there are some new variants of CRISPR (e.g., CRISPR-mediated prime editor, CRISPR-mediated base editor, CRISPR interference (CRISPRi), CRISPR activation (CRISPRa), and CRISPR-mediated epigenome editing) that can be used to produce novel ornamental cannabis varieties.

Knocking out protein-coding genes to improve ornamental traits of cannabis might be necessary but could prove to be problematic. Removing indispensable genes from the cannabis genome can lead to pleiotropic and/or lethal effects. Conditional CRISPR/Cas systems represent the best solution for tackling this obstacle. With the application of different tissue-specific promoters, Cas9 expression can be limited to particular cell types [87], restricting CRISPR-mediated genome engineering to specific tissues or organs (Figure 6). Recently, the conditional CRISPR/Cas system was successfully used to determine gene function in the stomatal lineage and root systems [88]. The methods described can thus be used to meet important objectives in developing ornamental cannabis lineages, including generating trichomes that produce only low levels of phytocannabinoids (i.e., specific organ) and manipulating growth habits or leaf color through genome editing.

## 4. Conclusions

Cannabis is a vigorous and fast-growing plant, valued for its aromatic inflorescences, green leaves, and beautiful leaflets. However, legal barriers and decades of stigmatization have largely limited its utility as an ornamental plant. Here, we have analyzed multifaceted commercial criteria at three levels (i.e., potential in ornamental sectors, sustainable exploitation feasibility, and readiness timescale for new value chain creation) as considerations for introducing cannabis as a new ornamental plant. Results indicate that cannabis has high ornamental potential as both a potted plant and landscaping plant, with high feasibility for sustainable exploitation. Furthermore, biochemical, physiological, morphological, and phenotypic variations among cannabis genotypes, along with tolerance to stressful conditions, can contribute to sustainable production of ornamental cannabis. However, the readiness timescale for ornamental cannabis production completely depends on the decriminalization and social attitude toward this plant in each country of interest. Cannabis chemotype Ⅴ, which produces virtually zero phytocannabinoids can be used as the foundation of ornamental cannabis research by overcoming certain legal barriers. Ultimately, mutagenesis and genome engineering-based approaches can facilitate cannabis’ introduction to the ornamental sector to further advance both horticultural and cannabis-related fields. Although the genome editing methods can be considered a promising approach for producing ornamental cannabis, genetically modified plants cannot be freely sold and cultivated in Europe, and thus the genome editing procedure described in this study is not currently possible for Europe.

## Figures and Tables

**Figure 1 plants-11-02383-f001:**
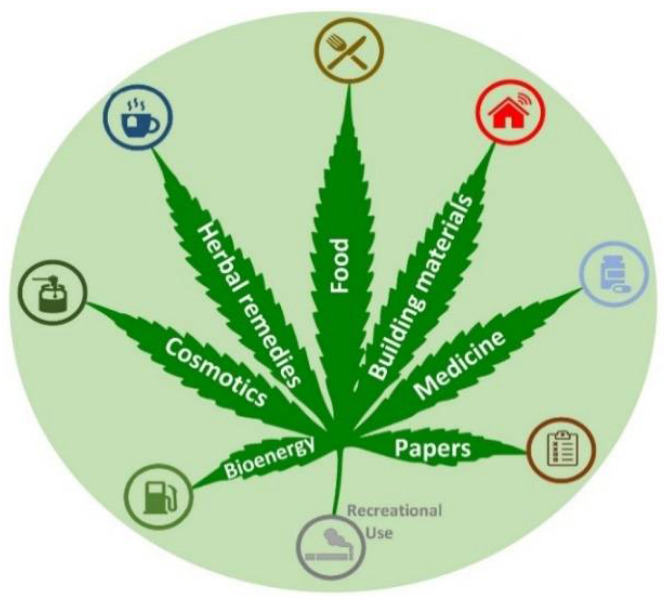
A schematic representation of different pharmacological and industrial applications of cannabis.

**Figure 2 plants-11-02383-f002:**
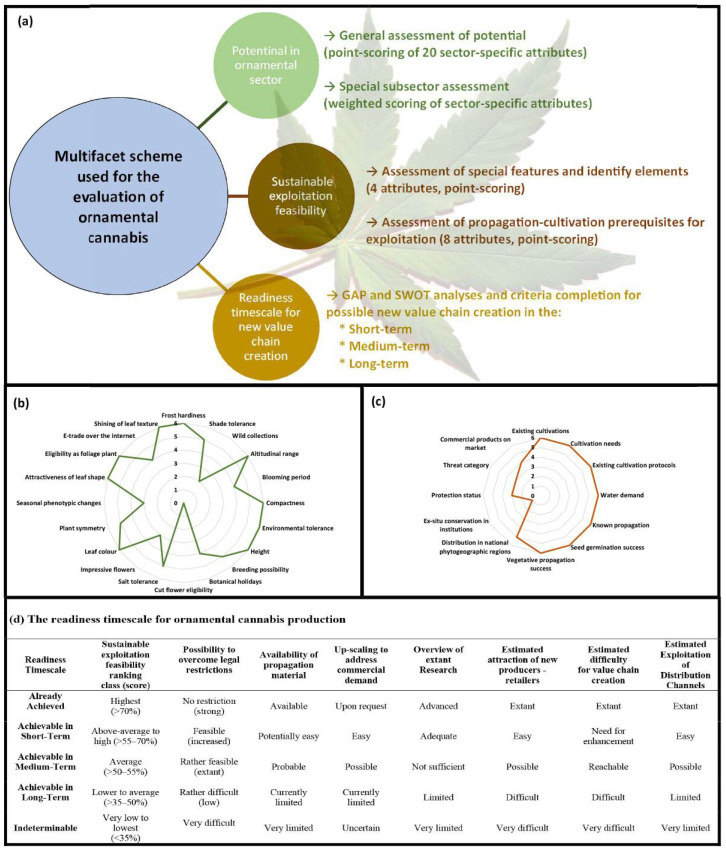
A schematic representation of the methodology for evaluating cannabis to introduce as an ornamental crop. (**a**) simplified schematic view of the multifaceted criteria at three levels for assessing cannabis in the ornamental industry, (**b**) evaluation of cannabis ornamental potential based on twenty attributes, (**c**) evaluation of cannabis sustainable exploitation feasibility based on twelve attributes, and (**d**) requirements of the readiness timescale for ornamental cannabis production.

**Figure 3 plants-11-02383-f003:**
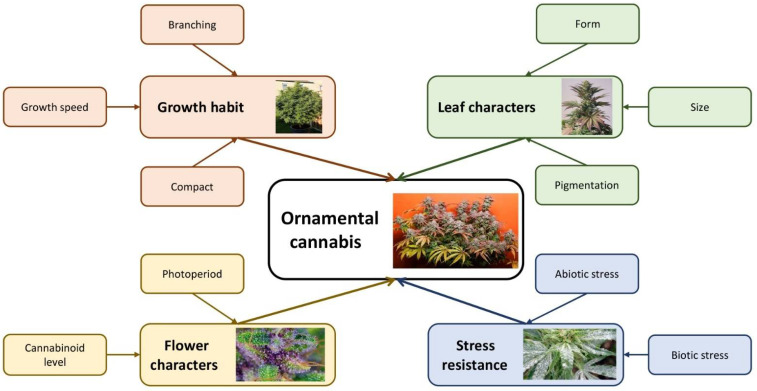
A schematic view of important economic traits of ornamental cannabis.

**Figure 4 plants-11-02383-f004:**
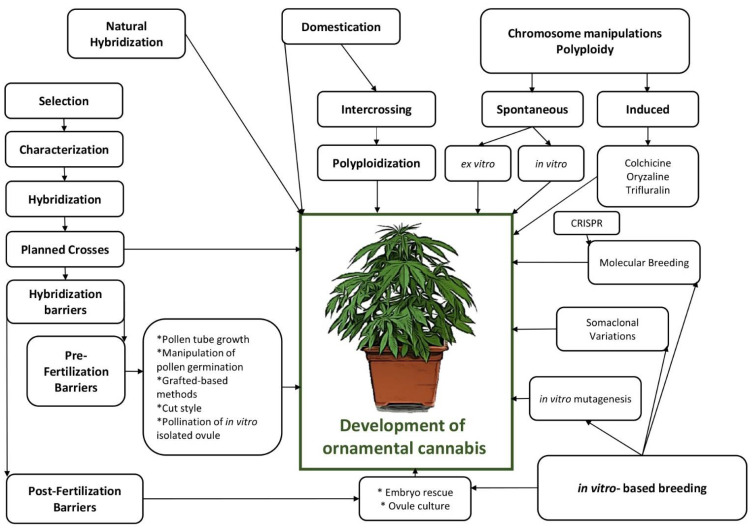
A schematic view of different breeding-based approaches for producing ornamental cannabis.

**Figure 5 plants-11-02383-f005:**
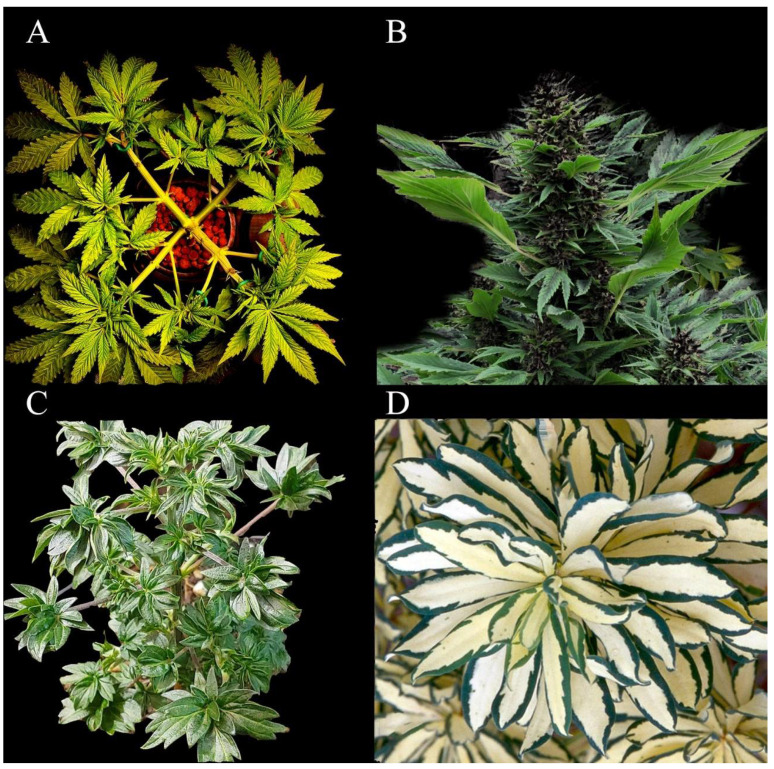
Some examples of cannabis mutants with ornamental value. (**A**) whorled phyllotaxy, (**B**) Ducksfoot, (**C**) Australian Bastard Cannabis, and (**D**) variegated leaves. Image credit: Dutch Passion.

**Figure 6 plants-11-02383-f006:**
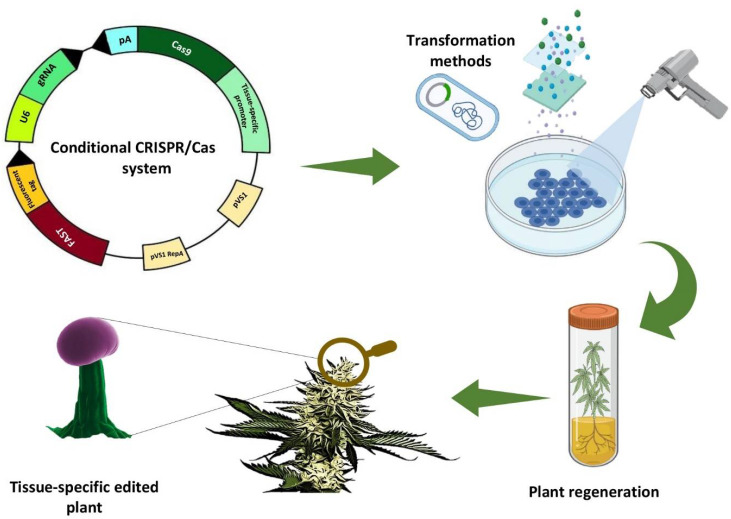
A schematic view of Conditional CRISPR/Cas methodology for producing ornamental cannabis.

## Data Availability

Not applicable.

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
