# Peer review of "New Insight into Ornamental Applications of Cannabis: Perspectives and Challenges"

_plants, 2022, doi:10.3390/plants11182383_

Round 1

Reviewer 1 Report

Dear Editor in Chief,

First and foremost, I would like to appreciate for providing me with the chance to review the draft titled “Ornamental Cannabis: Perspectives and Challenges” by Hesami and colleagues. In this perspective letter, the authors tried to open a new window to the application of cannabis as an ornamental plant, which due to the legal restriction this aspect of the mentioned plant has not been addressed well. In the first step of this perspective, the authors analyzed multifaceted commercial criteria and confirmed the high value of cannabis as an ornamental plant. Then, the authors tried to find the impact of factors on the application of the plant as an ornamental herb. Based on their assessment, the authors highlighted that “legal status” is one of the most important factors affecting the ornamental application of the plant. In the next step, the authors focused on the potential of cannabis chemotype â…¤, as the foundation for breeding ornamental traits through different plant biotechnology approaches. At the end of the current effort, the authors conclude that breeding and legalization approaches of this plant for ornamental utility can offer boundless opportunities related to economics and executive business branding. In general, I believe that this paper looks into the ornamental application of this plant from a new angle and it could be a good model for further studies of the same plant and other plants as well.  

I have found this paper interesting, and I would like to suggest the paper for publication after some minor corrections. In addition, the paper has been written and organized well, and the English language of the paper is acceptable.

Comments:

-       I would like to suggest authors change the title of this paper to “New insight to the Ornamental Application of Cannabis, Perspectives and Challenges”.

-       I also would like to suggest, that authors address some more ornamental characteristics of the plant either in the introduction or open a new sub-tittle, maybe under the third paragraph of the introduction.

-       Please double-check the spacing between the words.

-       Authors can remove figure 1, as it’s not informative.

-       In vitro should be italic.

Best wishes

Author Response

I have found this paper interesting, and I would like to suggest the paper for publication after some minor corrections. In addition, the paper has been written and organized well, and the English language of the paper is acceptable.

We thank you for your careful consideration of this manuscript and for giving us an opportunity to revise it. We appreciate your positive and constructive comments and suggestions. We have tried our best to answer all questions and revise the manuscript as suggested. The following are our detailed responses to your comments.

Comments:

-       I would like to suggest authors change the title of this paper to “New insight to the Ornamental Application of Cannabis, Perspectives and Challenges”.

Thanks for your great suggestion. The title was changed based on your suggestion.

-       I also would like to suggest, that authors address some more ornamental characteristics of the plant either in the introduction or open a new sub-tittle, maybe under the third paragraph of the introduction.

Ornamental characteristics of the plant have been added (lines 36-42).

-       Please double-check the spacing between the words.

We double-checked the spacing between the words.

-       Authors can remove figure 1, as it’s not informative.

Since Figure 1 shows the current application of cannabis, we believe that it can help readers to understand all the benefits of cannabis.

-       In vitro should be italic.

We amended it and right now in vitro is in italics.

Reviewer 2 Report

The perspective paper “Ornamental Cannabis: Perspectives and Challenges” is proposed the use of Cannabis sativa L. chemotype V in horticultural industry as ornamental plant and also presented the breeding methods can be used for this goal. According to my opinion the following has to be taken into consideration before the publishing

In introduction the writers mentioned the chemotype V cannabis, but only in page 7 (session 3), they explain what is chemotype V cannabis. An explanation is also needed to introduction.

In session 2, the point score used to evaluate the general ornamental potential of cannabis are based on literature data, which means that the writers took from literature all these  markers (frost hardiness, shade tolerance, wild collections, altitudinal range, blooming period, compactness, environmental tolerance, height, breeding possibility, botanical holidays, cut flower eligibility, salt tolerance, impressive flowers, leaf color, plant symmetry, seasonal phenotypic changes, attractiveness of leaf shape, eligibility as foliage plant, e-trade over the internet, and shining of leaf texture). It was not mentioned from which varieties and areas are these data because each variety is different from other to most of these characteristics and so the ornamental potential of cannabis can be changed according to variety. Also, which is the reference for the standard 67.5% score?

Session 2: The results of suitability as potted or landscaping plant from how many estimations are? I think that the way presented these results are inadequate, please rewrite this session.

In the last paragraph of session 2, I have to notice that the writers have to mention industrial hemp which is cultivated legally in most of the European countries, for varieties registered to European catalogue and their THC levels are <0.2%. 

Avoid using “we” in abstract and introduction session.

Author Response

* The perspective paper “Ornamental Cannabis: Perspectives and Challenges” is proposed the use of Cannabis sativa L. chemotype V in horticultural industry as ornamental plant and also presented the breeding methods can be used for this goal. According to my opinion the following has to be taken into consideration before the publishing

We thank you for your careful consideration of this manuscript and for giving us an opportunity to revise it. We appreciate your constructive comments and suggestions. We have tried our best to answer all questions and revise the manuscript as suggested. The following are our detailed responses to your comments.

* In introduction the writers mentioned the chemotype V cannabis, but only in page 7 (session 3), they explain what is chemotype V cannabis. An explanation is also needed to introduction.

The explanation of chemotype V (i.e. an undetectable amount of phytocannabinoids (almost zero phytocannabinoids)) was added to the introduction section (lines 63-64).

* In session 2, the point score used to evaluate the general ornamental potential of cannabis are based on literature data, which means that the writers took from literature all these  markers (frost hardiness, shade tolerance, wild collections, altitudinal range, blooming period, compactness, environmental tolerance, height, breeding possibility, botanical holidays, cut flower eligibility, salt tolerance, impressive flowers, leaf color, plant symmetry, seasonal phenotypic changes, attractiveness of leaf shape, eligibility as foliage plant, e-trade over the internet, and shining of leaf texture). It was not mentioned from which varieties and areas are these data because each variety is different from other to most of these characteristics and so the ornamental potential of cannabis can be changed according to variety. Also, which is the reference for the standard 67.5% score?

We largely cited review papers and/or textbooks that mentioned most of the specific markers. As we mentioned in the manuscript, In the current study, we evaluated the ornamental potential of cannabis by utilizing the procedure developed by Krigas, et al. (2021), which accounts for all subsectors of the ornamental industry (e.g., cut-flowers, pot plants, bedding plants) (please see lines 109-111). In fact, this procedure evaluates the ornamental potential of plants based on the availability of information and important traits. If a specific cultivar contains the traits (e.g., attractiveness of leaf shape, salt tolerance, etc.) it can be used in further breeding programs to produce ornamental plants. Please consider that this section of the procedure only evaluates the ornamental POTENTIAL of the plant (i.e., species not cultivar).

Also, the reference for the standard 67.5% score is Krigas, et al. (2021) who developed this procedure for the horticultural industry. We cited this reference at the end of the sentence (line125).

* Session 2: The results of suitability as potted or landscaping plant from how many estimations are? I think that the way presented these results are inadequate, please rewrite this session.

The suitability as potted and landscaping plant are subsections of the general ornamental potential of the plant. So, all the markers (estimations) are exactly the same as they are used for general ornamental potential. For instance, in the procedure for evaluating the suitability as potted plant, 9 markers out of 20 markers of the general ornamental potential of the plants have been used. Both procedures for suitability as potted and landscaping plant is previously developed by Krigas, et al. (2021) for using in the horticultural industry.

* In the last paragraph of session 2, I have to notice that the writers have to mention industrial hemp which is cultivated legally in most of the European countries, for varieties registered to European catalogue and their THC levels are <0.2%.

Thanks for your great suggestion. We added this sentence to the manuscript based on your suggestion (lines 227-229).

* Avoid using “we” in abstract and introduction.

We amended the mentioned sentences in abstract and introduction (lines 12-14; 18; 98-99)

Round 2

Reviewer 2 Report

The perspective paper “Ornamental Cannabis: Perspectives and Challenges” which title change to "New Insight into Ornamental Application of Cannabis: Perspectives and Challenges" is accepted for publication in current form as given a new approach to cannabis plants.

Author Response

Thanks